# Adaptive State-of-Charge Estimation for Lithium-Ion Batteries by Considering Capacity Degradation

**Peipei Xu, Junqiu Li \*, Chao Sun** **, Guodong Yang and Fengchun Sun**

National Engineering Laboratory for Electric Vehicles, School of Mechanical Engineering, Beijing Institute of Technology, Beijing 100081, China; xup_chd@163.com (P.X.); chaosun@bit.edu.cn (C.S.); yangguodpmng@bit.edu.cn (G.Y.); sunfch@bit.edu.cn (F.S.)
**\*** Correspondence: lijunqiu@bit.edu.cn

**Abstract:** The accurate estimation of a lithium-ion battery's state of charge (SOC) plays an important role in the operational safety and driving mileage improvement of electrical vehicles (EVs). The Adaptive Extended Kalman filter (AEKF) estimator is commonly used to estimate SOC; however, this method relies on the precise estimation of the battery's model parameters and capacity. Furthermore, the actual capacity and battery parameters change in real time with the aging of the batteries. Therefore, to eliminate the influence of above-mentioned factors on SOC estimation, the main contributions of this paper are as follows: (1) the equivalent circuit model (ECM) is presented, and the parameter identification of ECM is performed by using the forgetting-factor recursive-least-squares (FFRLS) method; (2) the sensitivity of battery SOC estimation to capacity degradation is analyzed to prove the importance of considering capacity degradation in SOC estimation; and (3) the capacity degradation model is proposed to perform the battery capacity prediction online. Furthermore, an online adaptive SOC estimator based on capacity degradation is proposed to improve the robustness of the AEKF algorithm. Experimental results show that the maximum error of SOC estimation is less than 1.3%.

**Keywords:** state of charge (SOC); equivalent circuit model (ECM); capacity degradation model; forgetting factor recursive least squares (FFRLS)

## 1. Introduction

Lithium-ion batteries (LIBs), with their high energy density, low pollution and low self-discharge rate, have become one of the main energy sources of electric vehicles (EVs) [1,2]. The accuracy and reliability of battery management system (BMS) can ensure the safety of EVs during driving. The accurate estimation of state of charge (SOC) and state of health (SOH) can improve battery life and utilization, which is very important to ensure system performance and reliable operation [3,4]. Therefore, many algorithms for the accurate estimation of SOC have been actively promoted.

A variety of SOC estimation methods have been applied, including the ampere–time integral method [5], open-circuit voltage method [6], data-driven methods [7] and model-based methods [8]. These algorithms have greatly improved the estimation of SOC. The ampere–hour integration method and the open circuit voltage (OCV) method are widely used in SOC estimation. The ampere–hour integral method is easily applicable to online SOC estimation; however, there are some errors in the current value due to the measurement errors during battery charging and discharging. As time progresses, the accumulated error will cause the SOC estimation accuracy to decrease continuously. OCV estimation is used to estimate SOC according to the mapping relationship between SOC and OCV. However, to obtain a stable OCV, it is necessary to withstand long-term static to eliminate the influence of the polarization effect, so it is not suitable for online SOC estimation. The data-driven methods do not reflect the reaction mechanism inside the battery. A black box model is used to describe the nonlinear relationship between SOC and its influencing factors. The

neural network (NN) method is one of the most widely applied data-driven algorithms at present. The author used NN to estimate SOC in [9,10] and obtained good estimation results. J. N et al. [11] established an SOC estimation model by using the support vector machine (SVM). Through simulation comparison, it was proved that the SVM has better robustness than NN, but NN and SVM need a great deal of experimental data to train the model, which increases the calculation burden. To reduce the estimation error of SOC, many model-based methods have been studied, including equivalent circuit models (ECMs) and electrochemical models. Kandler et al. [12] established a simplified electrochemical model of a power battery and completed the estimation of the SOC, but the parameter acquisition was complex. At present, the ECM has been extensively used for the BMS of EVs. Ye et al. [13] proposed particle swarm optimization to optimize the Extended Kalman filter (EKF) to estimate SOC; Xiong et al. [14] used the multi-scale EKF to realize the joint estimation of the parameters and states of LIBs. Although EKF can effectively obtain good estimation results, it ignores the higher-order term of the Taylor series expansion of the nonlinear function, which significantly reduces the estimation accuracy of SOC. Therefore, the Unscented Kalman filter (UKF) is proposed, which is based on unscented transformation. He et al. [15] used UKF to estimate SOC and showed that UKF has better estimation accuracy than EKF. Nejad S. et al. [16] proposed the Adaptive Extended Kalman filter (AEKF) to obtain a better estimation result, and the SOC estimation error was less than 2%. Although the above methods can achieve better estimation results, capacity degradation is not considered.

Although ECMs have achieved some progress in battery modeling and SOC estimation, to promote the accuracy of SOC estimation, the core contributions of this work are as follows: (1) the ECM is established as the battery model in this paper, and the parameters are identified by the forgetting-factor recursive-least-squares (FFRLS) method; (2) the battery capacity degradation model is proposed; (3) a new battery model based on AEKF combined with capacity degradation is proposed to promote the accuracy of SOC estimation; and (4) the accuracy of the model is verified under driving conditions.

The outline of the paper is as follows. The introduction is presented in Section 1. Section 2 describes the fundamental battery model and parameter identification method. In Section 3, the AEKF method based on the online parameter identification method is presented, and the sensitivity analysis of SOC estimation to capacity degradation is presented in Section 4. The effectiveness of the proposed model is verified in Section 5, and the conclusions are shown in Section 6.

## 2. Battery Model and Parameter Identification

The dynamic voltage characteristics of lithium-ion batteries show mutagenicity and gradualness. The ECM selected in this paper is a second-order equivalent circuit model, as shown in Figure 1.

$$\begin{cases} \overset{\bullet}{U_d} = -\frac{U_d}{C_d R_d} + \frac{I_L}{C_d} \\ \overset{\bullet}{U_c} = -\frac{U_c}{C_c R_c} + \frac{I_L}{C_c} \\ U_t = U_{oc} - U_d - U_c - R_0 I_L \end{cases} \tag{1}$$

where $U_d$, $U_c$ is the $R_d$, $R_c$ two-terminal voltage, respectively, $I_L$ is the charge and discharge current of the battery module, $U_t$ is the battery's terminal voltage and $U_{oc}$ is the open circuit voltage (OCV).

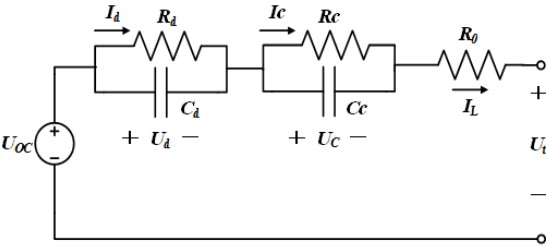

**Figure 1.** Equivalent circuit model (ECM).

In order to realize the online estimation of system parameters, we use the recursive least squares (RLS) method. However, the parameters of the battery system change slowly, so the algorithm finds it difficult to obtain accurate parameters.

Therefore, adding the forgetting factor to the recursive least squares method can effectively solve this problem and realize the online estimation of battery parameters. The flowchart of the FFRLS algorithm is shown in Figure 2, and the algorithm is as follows:

$$y_k = \Phi_k^T \theta_k + e_k \tag{2}$$

where $e_k$ is the zero-mean white noise, $\theta_k$ is the parameter matrix and $\Phi_k$ is the data matrix.

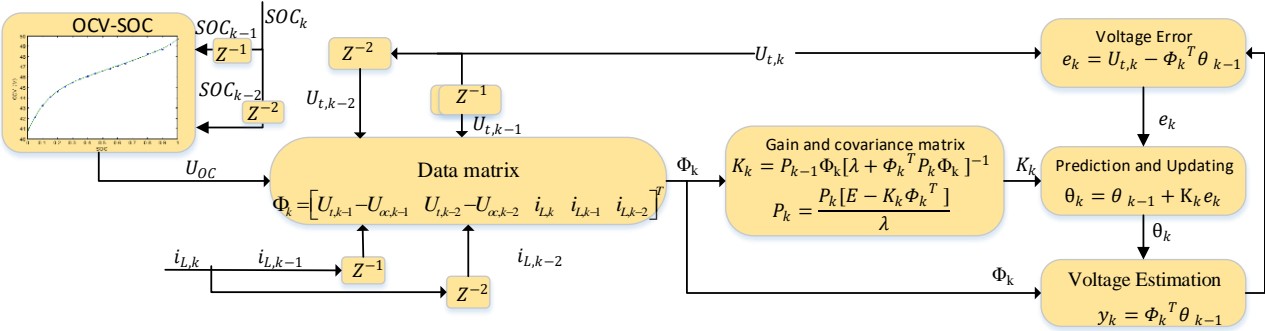

**Figure 2.** Battery parameter identification based on the forgetting-factor recursive-least-squares (FFRLS) method. OCV: open circuit voltage; SOC: state of charge.

In order to obtain the parameters of the ECM, the state of the model is transformed into a mathematical form which can be identified by the RLS method:

$$\frac{U_t(s) - U_{oc}(s)}{I_L(s)} = \frac{U_{rc}(s)}{I_L(s)} = -\left( R_0 + \frac{R_c}{1 + R_c C_c s} + \frac{R_d}{1 + R_d C_d s} \right) \tag{3}$$

We use the Euler algorithm to discretize this, which is defined in Equation (4):

$$s = \frac{2}{\omega} \frac{z - 1}{z + 1} \tag{4}$$

where $\omega$ is the sampling time interval, and Equation (3) can be transformed as follows:

$$\frac{U_{rc}(z)}{I_L(z)} = \frac{a_3 z^2 + a_4 z + a_5}{z^2 - a_1 z - a_2} \tag{5}$$

where

$$\begin{cases} a_1 = -\dfrac{2\omega^2 - 8R_cC_cR_dC_d}{\omega^2 + 2\omega(R_cC_c + R_dC_d) + 4R_cC_cR_dC_d} \\[3mm] a_2 = -\dfrac{\omega^2 - 2\omega(R_cC_c + R_dC_d) + 4R_cC_cR_dC_d}{\omega^2 + 2\omega(R_cC_c + R_dC_d) + 4R_cC_cR_dC_d} \\[3mm] a_3 = -\dfrac{\omega^2(R_0 + R_c + R_d) + 2\omega(R_0R_cC_c + R_0R_dC_d + R_cR_dC_d + R_dR_cC_c) + 4R_0R_cC_cR_dC_d}{\omega^2 + 2\omega(R_cC_c + R_dC_d) + 4R_cC_cR_dC_d} \\[3mm] a_4 = -\dfrac{2\omega^2(R_0 + R_c + R_d) - 8R_0R_cC_cR_dC_d}{\omega^2 + 2\omega(R_cC_c + R_dC_d) + 4R_cC_cR_dC_d} \\[3mm] a_5 = -\dfrac{\omega^2(R_0 + R_c + R_d) - 2\omega(R_0R_cC_c + R_0R_dC_d + R_cR_dC_d + R_dR_cC_c) + 4R_0R_cC_cR_dC_d}{\omega^2 + 2\omega(R_cC_c + R_dC_d) + 4R_cC_cR_dC_d} \end{cases} \quad (6)$$

We perform a *z* inverse transform on Equation (5) to obtain Equation (7):

$$U_k = (1 - a_1 - a_2)U_k + a_1 U_{k-1} + a_2 U_{k-2} + a_3 I_{L,k} + a_4 I_{L,k-1} + a_5 I_{L,k-2} \quad (7)$$

Since the sampling time T is very small, $U_{oc}$ is almost unchanged; that is,

$$U_{oc,k} - U_{oc,k-1} \approx 0$$

Therefore, Equation (7) can be simplified as

$$U_k = (1 - a_1 - a_2)U_k + a_1 U_{k-1} + a_2 U_{k-2} + a_3 I_{L,k} + a_4 I_{L,k-1} + a_5 I_{L,k-2} \quad (8)$$

The output matrix $y_k$, parameter matrix $\theta$ and data matrix $\Phi$ can be achieved:

$$\begin{cases} y_k = U_k = \Phi_k^T \theta_k \\ \theta_k = \begin{bmatrix} a_1 & a_2 & a_3 & a_4 & a_5 \end{bmatrix}^T \\ \Phi_k = \begin{bmatrix} U_{k-1} - U_{oc,k-1} & U_{k-2} - U_{oc,k-2} & i_{L,k} & i_{L,k-1} & i_{L,k-2} \end{bmatrix}^T \end{cases} \quad (9)$$

Using the FFRL method to calculate $\theta$, the parameters of the model can be obtained with Equation (6):

$$\begin{cases} R_0 = \dfrac{-a_3 + a_4 - a_5}{1 + a_1 - a_2} \\[2mm] R_0 + R_c + R_d = \dfrac{-a_3 - a_4 - a_5}{1 - a_1 - a_2} \\[2mm] R_cC_cR_dC_d = \dfrac{\omega^2(1 + a_1 - a_2)}{2(1 - a_1 - a_2)} \\[2mm] R_cC_c + R_dC_d = \dfrac{\omega(1 + a_2)}{1 - a_1 - a_2} \\[2mm] R_0R_cC_c + R_0R_dC_d + R_cR_dC_d + R_dR_cC_c = \dfrac{\omega(a_5 - a_3)}{1 - a_1 - a_2} \end{cases} \quad (10)$$

## 3. AEKF-Based SOC Estimation

In order to estimate SOC accurately, the AEKF algorithm with time-varying statistical characteristics is adopted in this chapter. Compared with the traditional Kalman filter, the AEKF algorithm takes innovation adaptive estimation as the core method, which can adaptively correct the system noise covariance and the measurement noise covariance. A discrete state space equation, which reflects the change of state variables such as SOC and voltage, is established.

$$\begin{bmatrix} SOC(k) \\ U_{R_cC_c}(k) \\ U_{R_dC_d}(k) \end{bmatrix} = \begin{pmatrix} 1 & 0 & 0 \\ 0 & \exp(\frac{-\omega}{\tau_c}) & 0 \\ 0 & 0 & \exp(\frac{-\omega}{\tau_d}) \end{pmatrix} \times \begin{bmatrix} SOC(k-1) \\ U_{R_cC_c}(k-1) \\ U_{R_dC_d}(k-1) \end{bmatrix} + \begin{bmatrix} \frac{-\eta\omega}{C} \\ R_c(1 - \exp(\frac{-\omega}{\tau_c})) \\ R_d(1 - \exp(\frac{-\omega}{\tau_d})) \end{bmatrix} \times i(k-1) + w(k-1) \quad (11)$$

where C is the current battery capacity, $\omega$ is the sampling period, $\tau_c$ and $\tau_d$ are the time constants of two RC loops, $\tau_c = R_cC_c$, $\tau_d = R_dC_d$ and $i(k-1)$ is current of a sample point

at time $k-1$. The discharge is positive and the charge is negative; $w(k-1)$ is the system process noise.

For any nonlinear discrete system, $f(x_k, u_k)$ is the system state equation and $h(x_k, u_k)$ is the observation equation of the system:

$$\begin{cases} x_{k+1} = f(x_k, u_k) + \omega_k \\ y_k = h(x_k, u_k) + v_k \end{cases} \tag{12}$$

where $x$ is the n-dimensional state vector, $u$ is the r-dimensional input vector, $y$ is the m-dimensional observed vector, $v_k$ is the observed noise and $\omega_k$ is the system noise, assuming that the noise mean is 0, the covariance is $R_k$ and $Q_k$, respectively, and $\omega_k$ and $v_k$ are mutually independent.

The equation of the linearized model is as follows:

$$\begin{cases} x_{k+1} \approx A_k x_k + [f(\hat{x}_k, u_k) - A_k \hat{x}_k] + \omega_k = A_k x_k + B_k u_k + \omega_k \\ y_k \approx C_k x_k + [h(\hat{x}_k, u_k) - C_k \hat{x}_k] + v_k = C_k x_k + D_k u_k + v_k \end{cases} \tag{13}$$

where

$$\begin{aligned} A_k &= \left. \frac{\partial f(x_k, u_k)}{\partial x_k} \right|_{x_k = \hat{x}_k} = \begin{bmatrix} 1 & 0 & 0 \\ 0 & \exp(-\omega/\tau_{d,k}) & 0 \\ 0 & 0 & \exp(-\omega/\tau_{c,k}) \end{bmatrix} \\ B_k &= \begin{bmatrix} -\frac{\eta\omega}{C_a} & R_{d,k}(1 - \exp(-\omega/\tau_{d,k})) & R_{c,k}(1 - \exp(-\omega/\tau_{c,k})) \end{bmatrix}^T \\ C_k &= \left. \frac{\partial g(x_k, u_k)}{\partial x_k} \right|_{x_k = \hat{x}_k} = \begin{bmatrix} \left. \frac{\partial U_{oc}}{\partial z} \right|_{z=z_k} & -1 & -1 \end{bmatrix} \\ D_k &= -R_{0,k} \end{aligned} \tag{14}$$

The flow chart of the SOC estimation algorithm based on FFRLS and AEKF is shown in Figure 3.

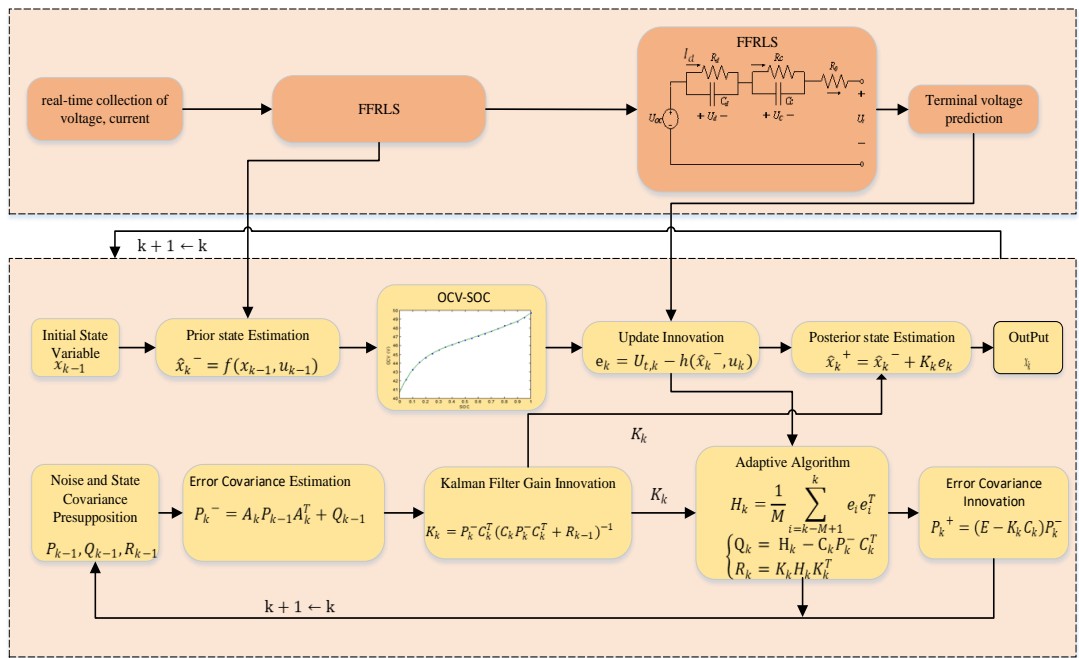

**Figure 3.** Flowchart of the SOC estimation algorithm.

*Sensitivity Analysis of SOC Estimation to Capacity Degradation*

The battery capacity will decrease along with the time of using the battery, which is an important variable for SOC estimation. To analyze the effects of the degradation of capacity in SOC estimation theoretically, we established a capacity error model to illustrate the sensitivity of SOC estimation to capacity degradation:

$$soc = soc_0 - \frac{1}{C_0} \int_0^t \eta \times I dt \tag{15}$$

where $C_0$ is the battery rated capacity, $I$ is the battery current, $\eta$ is the charge discharge efficient and $soc_0$ is assumed to be 1.

$$soc^* = soc_0 - \frac{1}{C_*} \int_0^t \eta \times I dt \tag{16}$$

where $C^*$ is the capacity under the specified condition and $SOC^*$ is the remaining capacity under the specified condition.

$$r_{soc} = SOC^* - SOC = \frac{(C^* - C_0) \int_0^t \eta I dt}{C_0 C^*} \tag{17}$$

where $r_{soc}$ is the absolute error in calculating SOC by the ampere–hour integral method.

From the above error analysis, we can see that the variation of capacity will have a great influence on SOC estimation. Therefore, establishing the capacity degradation model is essential for SOC estimation. Meanwhile, the capacity, which was obtained from the accelerated life test, cannot meet the requirement because of the complex and varying situation. Therefore, whether the battery capacity can be accurately estimated in real time is related to the accuracy of SOC estimation.

## 4. Adaptive SOC Estimator Based on Degradation Model

*4.1. SOC Estimator Based on Degradation Model*

This section focuses on the prediction method of the maximum available capacity and the process of establishing the capacity degradation model. After comparative analysis, the capacity degradation model under dynamic conditions is selected to realize the updating of the available capacity. If accumulative error can be reduced, the prediction accuracy can be improved. Furthermore, the real-time capacity prediction can be added to the AEKF algorithm to improve the robustness of the SOC estimation effectively. The flow chart of the SOC estimation algorithm based on updating the battery capacity is shown in Figure 4.

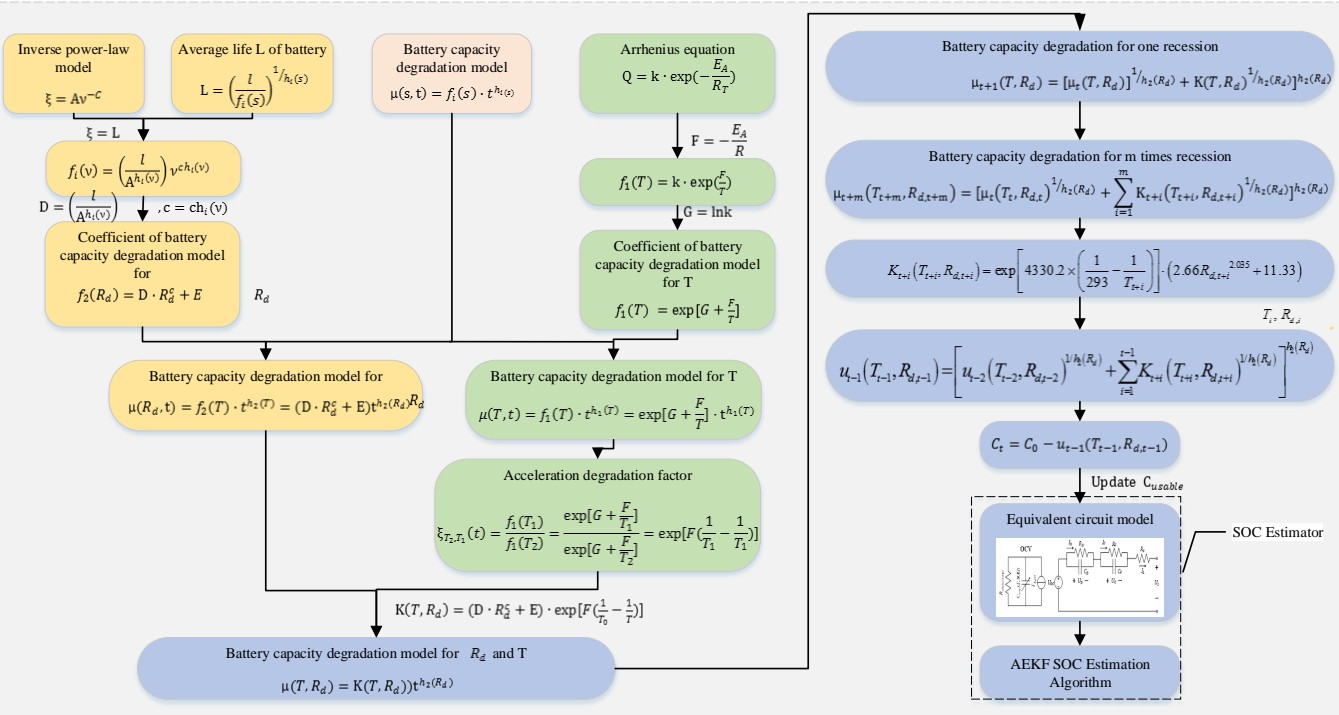

**Figure 4.** SOC estimator based on updating the battery capacity. AEKF: Adaptive Extended Kalman filter.

### 4.2. Lithium-Ion Battery Degradation Model

When the battery is charged/discharged, an LIB will lose its capacity $\Delta C$. Because of the complexity of the environment and the internal state of the LIB during each cycle, $\Delta C$ is independent and identically distributed, which is a random variable. Therefore, after $t$ cycles, the accumulative capacity of the battery is degraded. $x(s,t)$ is the sum of $\Delta C$ per cycle. According to the central limit theorem, the capacity degradation of the battery under stress obeys the normal distribution:

$$x(s,t) \sim N\left(u(s,t), \sigma^2(n)\right) \tag{18}$$

where $x(s,t)$ is the cumulative capacity recession of the battery after $t$ cycles under $s$ stress level, $u(s,t)$ is the mean value of the capacity recession after $t$ times cycle under $s$ stress level and $\sigma^2(t)$ is the capacity degradation variance after $t$ cycles under $s$ stress level.

Thus, the capacity degradation equation of battery power is as follows:

$$u(s,t) = f_i(s) \times t^{h_i(s)} \tag{19}$$

where $s$ is the stress type, $h_i(s)$ is constant, $s$ can be the temperature $T$ and discharge rate $R_d$ and $i = 1, 2$.

#### 4.2.1. Lithium-Ion Battery Degradation Model under Static Conditions

As we assume that the charging mode of the vehicle is fixed and the charge rate is constant at a small rate, the effect of the charge rate on the capacity degradation of the lithium-ion battery pack is not considered in this paper. The following models of cell capacity degradation under the conditions of temperature and discharge rate are analyzed and studied based on the above theory.

(1)  Capacity degradation model under constant temperature

The coefficient of the decay function is as follows:

$$f_1(T) = K \exp\left(\frac{F}{T}\right) \tag{20}$$

where $K$ is the fitting coefficient, $F$ is the pressure and $T$ is the temperature.

Assuming Equation (20) can be transformed to

$$f_1(T) = \exp\left[G + \frac{F}{T}\right], \tag{21}$$

The capacity recession model under the condition of temperature stress is obtained by Equation (21):

$$u(T,t) = f_1(T) \times t^{h_1(T)} = \exp\left[G + \frac{F}{T}\right] \times t^{h_1(T)} \tag{22}$$

The parameters $G$, $F$, $h_1(T)$ are obtained by fitting the capacity degradation data under the temperature parameter.

The capacity degradation of the battery at different temperatures and different charging and discharging rates is shown in Table 1. The fitting results of the capacity degradation during 300 cycles under different temperature stresses are shown in Figure 5.

**Table 1.** Capacity degradation under different conditions.

| Number of Cycles | | 1 | 50 | 100 | 150 | 200 | 250 | 300 |
|---|---|---|---|---|---|---|---|---|
| Temperature | 20 °C | 35.37 Ah | 35.06 Ah | 34.84 Ah | 34.64 Ah | 34.45 Ah | 34.27 Ah | 34.10 Ah |
| | 40 °C | 35.23 Ah | 34.44 Ah | 33.86 Ah | 33.35 Ah | 32.87 Ah | 32.41 Ah | 31.97 Ah |
| Discharge rate | 1C | 35.37 Ah | 35.06 Ah | 34.84 Ah | 34.64 Ah | 34.45 Ah | 34.27 Ah | 34.10 Ah |
| | 2C | 35.87 Ah | 35.38 Ah | 35.02 Ah | 34.71 Ah | 34.41 Ah | 34.13 Ah | 33.86 Ah |
| | 3C | 35.59 Ah | 34.79 Ah | 34.21 Ah | 33.69 Ah | 33.21 Ah | 32.75 Ah | 32.31 Ah |

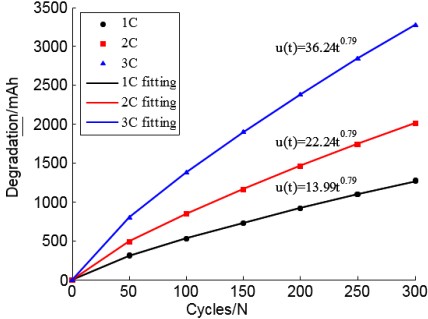

**Figure 5.** Capacity degradation curves at different discharge rates.

Based on the battery capacity degradation equation under temperature stress and the results from Figure 6, the following equation can be obtained:

$$\begin{cases} \exp\left(G + \frac{F}{293}\right) = 13.99 \\ \exp\left(G + \frac{F}{313}\right) = 35.97 \end{cases} \tag{23}$$

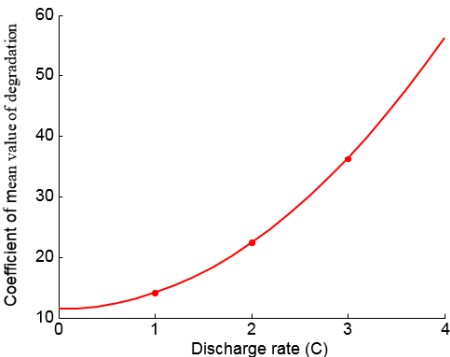

**Figure 6.** Recession parameter curves at different discharge rates.

Therefore, $F = -4330.2$, $G = 17.4$ and the degradation equation of the battery under temperature stress is

$$u(T, t) = \exp\left[17.4 - \frac{4330.2}{T}\right] \times t^{0.79} \tag{24}$$

(2)　Battery capacity model under constant discharge rate

$$L = \left(\frac{l}{f_i(s)}\right)^{1/h_i(s)} \tag{25}$$

From Equation (19), the capacity degradation model under the $R_d$ is

$$u(R_d, t) = \left(D \times R_d^{c'} + E\right) \times t^{h_2(R_d)} \tag{26}$$

where $D$, $c'$, $E$, $h_2(R_d)$ are the constants; these can be fitted by the data of battery capacity degradation under discharge rate parameters. According to the data in Table 1, the fitting curve and the degradation equation under different discharge rates and battery capacity degradation can be obtained and are shown in Figure 7, and the parameters of capacity degradation equation can be obtained and are shown in Table 2. After fitting the decay parameters of different discharge rates, the coefficient curve of the degradation equation can be obtained and is shown in Figure 6, the equation of which is shown in Equation (27):

$$f_2(R_d) = 2.66R_d^{2.035} + 11.33 \tag{27}$$

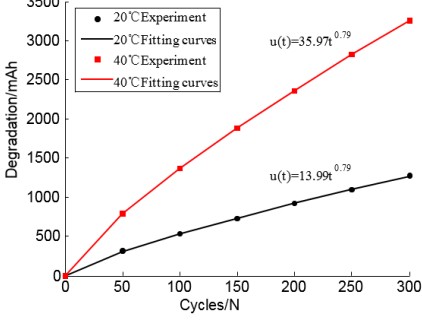

**Figure 7.** Fitting curves of capacity degradation at different temperatures.

**Table 2.** Identification parameters under the degenerating equation.

| C-Rate | $f_2(R_d)$ | $h_2(R_d)$ |
|:------:|:----------:|:----------:|
| 1C | 13.99 | 0.7935 |
| 2C | 22.24 | 0.7921 |
| 3C | 36.24 | 0.7927 |

From Equation (26), the degradation equation of battery modules at different discharge rates is obtained, as shown in Equation (28).

$$u(R_d, t) = f_2(R_d)t^{h_2(R_d)} = \left(2.66R_d^{2.035} + 11.33\right) \times t^{0.79} \tag{28}$$

(3) Capacity degradation under compound stress

Because a normally working battery will be affected by high-temperature stress and a high discharge rate at the same time, when the temperature rises, the capacity degradation will be accelerated. The capacity degradation model under discharge rate stress alone will not accurately predict this degradation. In this paper, the accelerated degradation factor of temperature stress is used to reflect the accelerated effect of temperature. Based on this, the capacity degradation model under combined stress is created.

The accelerated degradation factor of temperature stress is as follows:

$$\xi_{T_2, T_1}(t) = \frac{f_1(T_1)}{f_1(T_2)} = \frac{\exp\left[G + \frac{F}{T_1}\right]}{\exp\left[G + \frac{F}{T_2}\right]} = \exp\left[F\left(\frac{1}{T_1} - \frac{1}{T_2}\right)\right] \tag{29}$$

From Equation (29) and setting $T_1 = T_0$, $T_2 = T$, the battery capacity degradation under $(T, R_d)$ stress is as follows:

$$u(T, R_d) = \xi_{T_2, T_1}(t)u(R_d, t) = \exp\left[F\left(\frac{1}{T_0} - \frac{1}{T}\right)\right] \times \left(D \times R_d^{c'} + E\right) \times t^{h_2(R_d)} \tag{30}$$

where $\exp\left[F\left(\frac{1}{T_0} - \frac{1}{T}\right)\right] \times \left(D \times R_d^{c'} + E\right)$ described the speed of capacity degradation under $(T, R_d)$; if the state is fixed, it is a constant. The coefficient of degradation is as follows:

$$K(T, R_d) = \left(D \times R_d^{c'} + E\right) \times \exp\left[F\left(\frac{1}{T_0} - \frac{1}{T}\right)\right] \tag{31}$$

The capacity degradation model can be obtained from Equation (30):

$$u(T, R_d) = K(T, R_d) \times t^{h_2(R_d)} \tag{32}$$

The capacity's accelerated fading factor under temperature stress can be determined with Equations (28) and (30).

$$\xi_{T_2, T_1}(t) = \exp\left[4330.2 \times \left(\frac{1}{293} - \frac{1}{T}\right)\right] \tag{33}$$

The capacity degradation model can be obtained from Equations (30) and (33).

$$u(T, R_d) = \exp\left[4330.2 \times \left(\frac{1}{293} - \frac{1}{T}\right)\right] \times \left(2.66R_d^{2.035} + 11.33\right) \times t^{0.79} \tag{34}$$

4.2.2. Lithium-Ion Battery Degradation Model under Dynamic Conditions

The capacity fading process under constant stress deviates greatly from the actual capacity fading process of battery power in actual use. Because there are two disadvantages of the degradation model under static conditions, the most important point is that battery power is not in a constant working situation, and so it is necessary to increase the discharge current to meet the power demand of electric vehicles under the situation of capacity degradation and the internal resistance increasing. In addition, according to the parameters of the battery power, the discharge rate tends to increase, which greatly affects the degradation rate of the power battery. Therefore, the capacity degradation model of battery power under dynamic stress parameters should be established to adapt the complex

working situations, and this model should describe the actual capacity degradation process more accurately.

For the dynamic stress, because the temperature is a slowly changing parameter, battery temperature is collected every 10 min, the average of which is defined as $T_i$; because the discharge rate $R_{d,I}$ is a real-time variable parameter, the value of the battery discharge current is collected in real time, and the ratio of the average discharge current collected in this cycle to the battery's rated capacity is finally calculated as the $R_{d,I}$ value. Based on this, the capacity degradation model is created in this paper. $u_t(T, R_d)$ is the t times capacity degradation model and $u_{t+1}(T, R_d)$ is the $t + 1$ times capacity degradation model, which is defined as Equation (35).

$$\begin{cases} u_t(T, R_d) = K(T, R_d) \times t^{h_2(R_d)} \\ u_{t+1}(T, R_d) = K(T, R_d) \times (t+1)^{h_2(R_d)} \end{cases}$$
$$\Rightarrow u_{t+1}(T, R_d) = \left[ u_t(T, R_d)^{1/h_2(R_d)} + K(T, R_d)^{1/h_2(R_d)} \right]^{h_2(R_d)} \tag{35}$$

where $u_{t+1}(T, R_d)$ is the total amount of recession after a recession at the $(T, R_d)$, which is based on the last recession. Thus, $u_t(T, R_d)$ is the degradation of the battery that has occurred, and it is related to the stress level of the previous cycle, but the previous recession is not related to the $(T, R_d)$ of this cycle.

The deformation formula from Equation (35) is as follows:

$$u_{t+1}(T_{t+1}, R_{d,t+1})^{1/h_2(R_d)} = u_t(T_t, R_{d,t})^{1/h_2(R_d)} + K_{t+1}(T_{t+1}, R_{d,t+1})^{1/h_2(R_d)}$$
$$\Rightarrow u_{t+m}(T_{t+m}, R_{d,t+m})^{1/h_2(R_d)} = u_t(T_t, R_{d,t})^{1/h_2(R_d)} + \sum_{i=1}^{m} K_{t+i}(T_{t+i}, R_{d,t+i})^{1/h_2(R_d)} \tag{36}$$

where $u_t(T_t, R_{d,t})$ is the initial capacity degradation and $u_{t+m}(T_{t+m}, R_{d,t+m})$ is the capacity degradation after $m$ cycles.

According to the fitting coefficient of the accelerated life test, the degradation coefficient equation is as follows:

$$K_{t+i}(T_{t+i}, R_{d,t+i}) = \exp\left[ 4330.2 \times \left( \frac{1}{293} - \frac{1}{T_{t+i}} \right) \right] \times \left( 2.66 R_{d,t+i}{}^{2.035} + 11.33 \right) \tag{37}$$

## 5. Experiments and Results

### 5.1. Dataset of Battery

In this paper, a lithium battery module was selected as the test object, which contained 12 series power battery monomers. Before grouping, these power batteries were strictly screened to ensure the consistency of the available capacity and internal resistance. The battery test platform consisted of a Digatron EVT 500-500 battery test system, a host computer, a temperature box and a Fluke data recorder for battery data acquisition. The battery charging and discharging equipment was the EVT 500-500, developed by a German company to test the battery power of electric vehicles, and the accuracy of voltage measurement and current measurement can reach 0.005. Furthermore, the maximum charging and discharging current can reach 500 A and the maximum voltage is 500 V. The battery test system is programmed by the upper computer, and the thermostat is used to adjust the current temperature. In order to analyze the hybrid power characteristics more intuitively, a Fluke data recorder was used, which can collect multiple sets of data according to requirements, and it can convert complex data into intuitive graphics and tables. The parameters of the used LIBs are presented in Table 3.

**Table 3.** Lithium manganese oxide battery parameters.

| Parameter | Value |
|---|---|
| Rated capacity | 35 Ah |
| Charging cut-off voltage | 4.2 V |
| Discharging cut-off voltage | 3.0 V |
| Rated voltage | 3.7 V |
| Cathode material | $LiMn_2O_4$ |
| Internal resistance | $\leq 1.0$ m$\Omega$ |

Based on the test platform built above, the static capacity test, battery accelerated life test and open circuit voltage (OCV) test were developed and edited by the upper computer of the Digatron test system. The batteries were charged and discharged 300 times at different temperatures and discharging rates. The static capacity test result is shown in Table 4. Taking the average value of the three times capacity value as the maximum available capacity, Tables 1 and 2 present the accelerated life test data. The fitting results are presented in Figures 5–7. The capacity degradation was recorded and the degradation equation parameter was derived from the capacity degradation data at different discharging rates. The actual project recession was similar to the experimental recession. We could observe the non-linear characteristics of the battery recession in different conditions, and these experimental data could be used to establish a capacity degradation model and predict the actual application of the battery performance. The battery pack was placed in a constant temperature environment of 20 °C to obtain the UDDS (Urban Dynamometer Driving Schedule) loading profiles, as shown in Figure 8.

**Table 4.** Static capacity test results.

| Constant Volume | Available Capacity (Ah) |
|---|---|
| First | 30.52 |
| Second | 30.12 |
| Third | 30.23 |

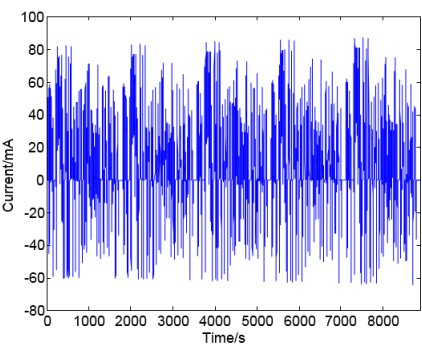

**Figure 8.** Current profiles of the battery for the Urban Dynamometer Driving Schedule (UDDS) test.

*5.2. SOC Estimation Results*

To verify the accuracy of the proposed model, the LIB experiment was carried out. The LIBs were placed in a 25 °C environment, loaded with UDDS cycles. From Figure 9, we can see that the maximum absolute error (MAE) between the estimated terminal voltage and the reference value was less than 2%, which can explain the high accuracy of voltage prediction. Thus, the accuracy of the identification of battery model parameters was increased.

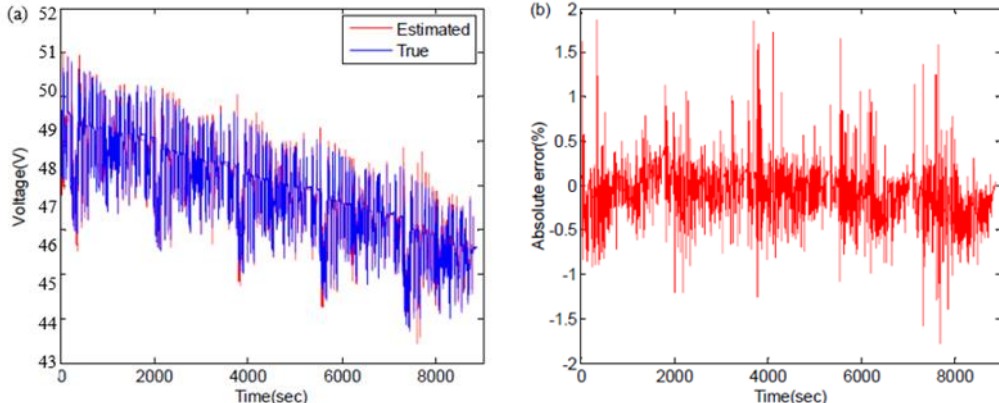

**Figure 9.** The terminal voltage estimation under UDDS cycles: (**a**) terminal voltage estimation; (**b**) terminal voltage estimation error.

From Figure 10, we can see that the capacity prediction results were very close to the actual capacity, and the maximum capacity error was 1.5% during the 600 full battery cycles. Therefore, this indicates that the capacity degradation model can describe the actual capacity fading process more accurately and it can update the maximum available capacity in real time and accurately.

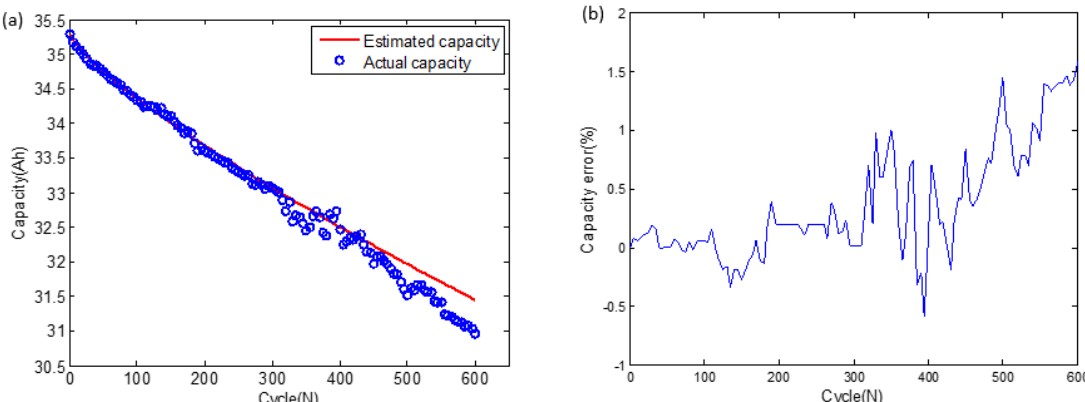

**Figure 10.** The battery capacity prediction result: (**a**) capacity prediction; (**b**) error of capacity prediction.

Figure 11 presents the result of SOC estimation in two states, and it shows that the absolute error of SOC estimation increases to about 6% if the capacity is not updated. After updating the capacity, the estimation error of SOC can be stabilized at about 0.5%. Therefore, it is necessary to incorporate the cell capacity degradation model into the repair procedure of the maximum available capacity in SOC estimation. Thus, when the capacity of batteries deteriorates, the capacity C of batteries should be updated in time to ensure the reliability of SOC estimation after battery aging.

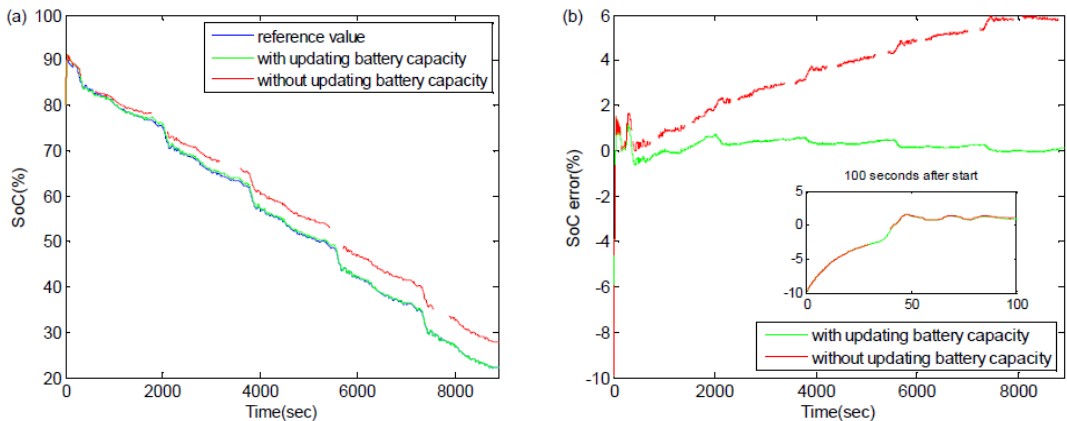

**Figure 11.** SOC estimation results in four states: (**a**) SOC estimation; (**b**) error of SOC estimation.

### 5.3. Hardware-in-the-Loop Validation

The BMS hardware-in-the-loop (HIL) simulation platform scheme designed in this paper is shown in Figure 12. In this scheme, the battery power in the form of a Simulink model was embedded into the SpeedGoat, a real-time simulation target. BMS communicated with the real-time simulation target machine through the Controller Area Network (CAN). The upper computer display terminal was programmed by LabVIEW, and the parameters were adjusted online and the data collected by Real-time Explorer of MATLAB software. The hardware in the loop simulation platform is shown in Figure 13. The BMS was provided with a 24 V voltage regulated power supply. The battery model was loaded into SpeedGoat by a Simulink compiler. BMS communicated with the simulator directly through CAN bus and was connected with the display terminal of the host computer through a USB-CAN card. The data were monitored in real time in the simulation process. The real-time simulation machine generated the virtual voltage and current as the virtual input of BMS. A background debug mode (BDM) background debugger was able to accomplish two functions including controller C code burning and online simulation debugging. When BMS was running, the registers and variables inside the BMS could be monitored online through the background debugger and online debugging functions of CodeWarrior software.

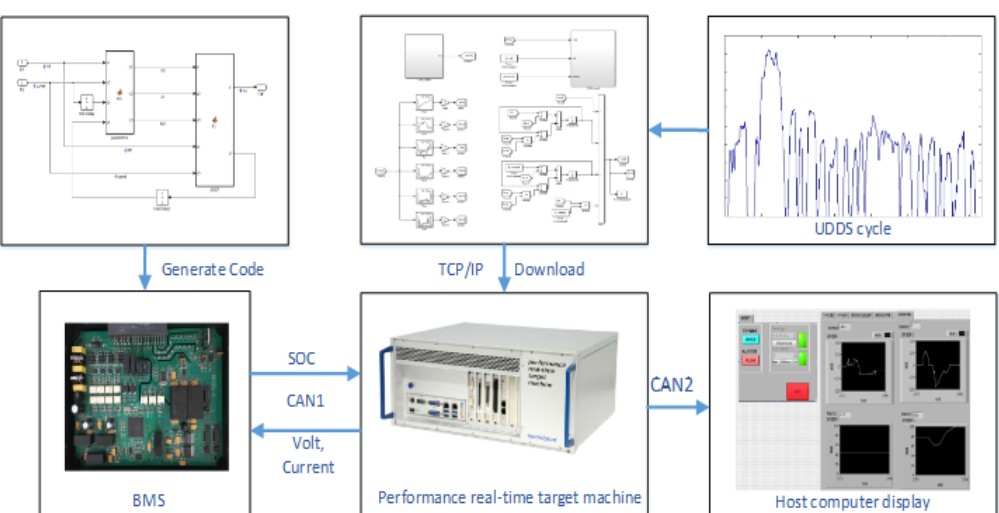

**Figure 12.** Hardware in the loop test scheme.

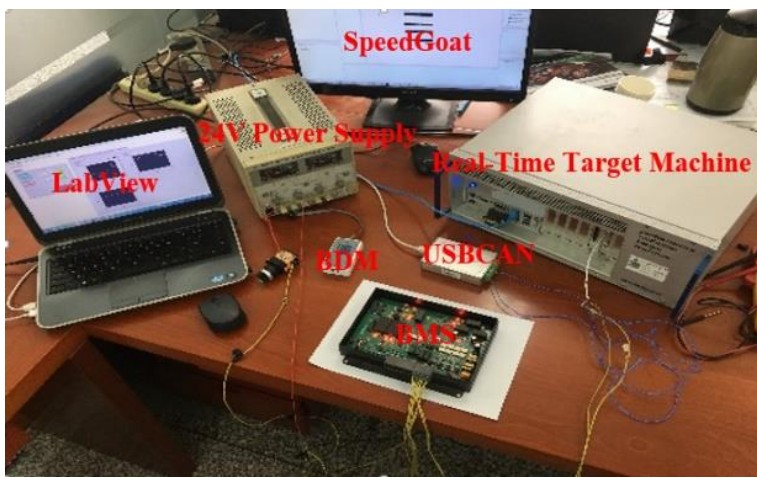

**Figure 13.** Hardware in the loop test platform.

Based on the HIL validation platform shown above, the UDDS operating condition data were imported into the real-time target simulator. The simulator calculated the current and voltage of the battery in real time, then sent the voltage and current to BMS by a CAN bus and obtained the SOC estimation result by BMS calculation. With the initial SOC value set as 80% and the actual SOC of the battery set as 90%, and running five UDDS cycles, the SOC estimation results and estimation errors are presented in Figure 14.

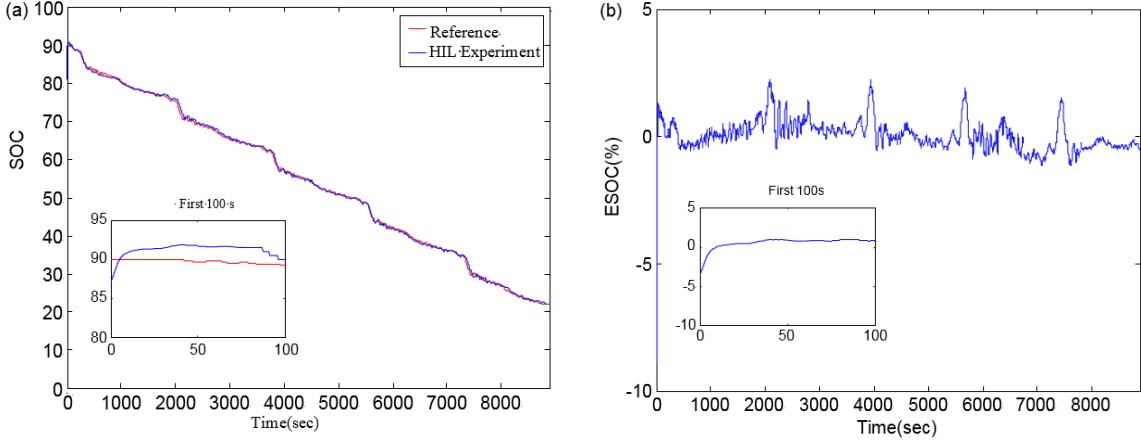

**Figure 14.** SOC estimation results: (**a**) SOC estimation; (**b**) the error of SOC estimation.

From Figure 14, we can see that the SOC estimation converged to the true value quickly at the beginning of the experiment. The SOC estimation error was less than 3% for the whole experiment. This illustrates that the SOC estimator considering capacity degradation achieves high accuracy. The trend of the SOC estimation curve was the same as that of the reference curve, but over time, the SOC estimation in the HIL test began to lag behind the reference value. We found that the SOC estimation was affected by the operation rate of the single-chip microcomputer, the acquisition rate of voltage and current and the data transmission rate. Therefore, when estimating the SOC of the battery, the capacity of the battery should be updated in real time when the battery capacity declines. At the same time, online parameter identification is necessary to ensure the accuracy of estimation.

## 6. Conclusions

In this paper, a battery equivalent circuit model was established, the RLS with the forgettable factor was adopted to realize parameter updating and the AEKF algorithm is used for SOC estimation. To improve the accuracy of the AEKF SOC estimator, the

sensitivity analysis of SOC estimation to capacity degradation was presented. Therefore, an online adaptive SOC estimator based on the capacity degradation was presented, and the battery capacity model under dynamic conditions was established, which could predict the battery capacity in real time. After 600 charge/discharge cycles, the experimental result shows that the capacity estimation error could be limited to 1.5%. Thus, the battery capacity prediction could meet the accuracy and real-time requirements. Meanwhile, the maximum error of the SOC estimator with capacity updates decreased from 6% to 1.23%. After verification in the real BMS controller, the calculation results indicate that the maximum error of the algorithm was less than 3%, thus meeting the requirements for on-board application. Therefore, the proposed method can not only improve SOC estimation accuracy but also can predict the battery capacity online.

**Author Contributions:** Conceptualization, P.X.; methodology, P.X. and J.L.; software, P.X. and C.S.; validation, P.X., C.S. and J.L.; formal analysis, G.Y.; investigation, G.Y. and C.S.; resources, P.X. and J.L.; writing—original draft preparation, P.X.; writing—review and editing, J.L. and C.S.; visualization, G.Y.; supervision, F.S.; project administration, C.S. and J.L.; funding acquisition, G.Y. and F.S. All authors have read and agreed to the published version of the manuscript.

**Funding:** The authors would like to express their thanks for the support of the Natural Science Foundation of China, Project: 5202291146.

**Informed Consent Statement:** Informed consent was obtained from all subjects involved in the study.

**Data Availability Statement:** No new data were created or analyzed in this study. Data sharing is not applicable to this article.

**Conflicts of Interest:** The authors declare no conflict of interest.

## Abbreviations

The following abbreviations are used in this manuscript:

| | |
|---|---|
| SOC | State of charge |
| EVs | Electrical vehicles |
| AEKF | Adaptive Extended Kalman filter |
| FFRLS | Forgetting factor recursive least squares |
| $K$ | Fitting coefficient |
| $F$ | Pressure |
| $T$ | Temperature |
| $t$ | Cycles |
| LIBs | Lithium-ion batteries |
| $R_{d,i}$ | Discharge rate |

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
