# Peer review of "Adaptive State-of-Charge Estimation for Lithium-Ion Batteries by Considering Capacity Degradation"

_electronics, doi:10.3390/electronics10020122_

Round 1
Reviewer 1 Report
In this manuscript an adaptive SoC estimation method for Li Ion Batteries were developed to optimize the accuracy of the SoC prediction in real time operation. This paper consider the aging of the battery and the change of the model parameter and capacity during operation. By establishing a equivalent circuit model and applying recursive least squares fitting method an online identification of model parameters are realized. Also with the battery capacity degradation model under dynamic conditions realization of better online capacity prediction could be shown. The presented work is new and show a high innovation.The autor could show an improved capacity estimation error limited till 1.5%. The model was realized in a BMS controller for verification and meets the on board application. The SOC estimation was improved and also the online battery capacitiy prediction. The introduction part is adequate described. The applied methods are explained and the theoretical back ground are well described.
There are some remarks:
Line 162: the Sympols should be better shown
Line 180: Please check k,F symbols. The Symbol T is given as Temperature and number of cycle. It is confusing.
Line 198 & 199: The sentence should be checked
Line 228: type error dynamic stress T,R:T?
For better reading the Tables numbers and explanations should be under the table.
Line 264: Typing error
Line 336: Figure 16 not shown
Line 368: The sentence was not finished
Author Response
Point 1: Line 162: the Symbols should be better shown 

Response 1: Thank you for your helpful suggestion. The size of the Figures 5 in this paper has been adjusted.
Point 2:Line 180: Please check K,F symbols. The Symbol T is given as Temperature and number of cycle. It is confusing.
Response 2: Thank you for your helpful suggestion. We redefine the meaning of each parameter as follows:
Where K is the fitting coefficient, F is the pressure, T is the temperature, and t is the number of cycles
Point 3:Line 198 & 199: The sentence should be checked
Response 3: I am sorry that this sentence was wrongly written in the original manuscript. And we have revised this sentence:
the parameters of capacity degradation equation can be obtained are shown in Table 4. And the decay curve of degradation equation are presented in Fig. 7.
Point 4:Line 228: type error dynamic stress T,R:T? For better reading the Tables numbers and explanations should be under the table.
Response4:I am sorry that this sentence was wrongly written in the original manuscript. And we have revised this sentence:
For the dynamic stress, because the temperature is a slowly changing parameter, so battery temperature is collected every 10 minutes and the average of which is defined as Ti; because discharge rate Rd,i is a real-time variable parameter, so the value of the battery discharge current is collected in real time, and the ratio of the average discharge current collected in this cycle to the battery rated capacity is finally calculated as the Rd,i value.
Point 5:Line 264: Typing error
Response 5:Table 4 and Table 5 are the accelerated life test data.
Point 6:Line 336: Figure 16 not shown
Response 6:I am sorry that this sentence was wrongly written in the original manuscript. Figure 16 has been changed to Figure 15
Point 7:Line 368: The sentence was not finished
Response 7:The authors would like to express their thanks for the support of Natural Science Foundation of China Project: 5202291146

Reviewer 2 Report
The present manuscript describes a SoC online estimation model which tries to mitigate the error from capacity degradation. The results show more accurate SoC estimation, with less than 3% error, and also online prediction of battery capacity.
The manuscript is, despite some typos, clearly written, results are well supported by evidence and properly described. The only thing this reviewer was not able to find is a list of terms used.
This reviewer would like to ask just one question, in addition to the comments made above:
- the results show the ability of the method to correct the SoC estimation and converge quickly to the true value. Authors mention that capacity of the battery should be updated in real time when the battery capacity declines. how would the method behave if, by any chance, a single cell of the battery pack dies after several tenths or hundreds of cycles, thus modifying the battery capacity?
From the point of view of this reviewer, the present manuscript can be published after language/typos correction.
Author Response
Response 1: Thank you for your helpful comment. The capacity of the battery decreases gradually when it is continuously cycled. The proposed method can update the battery capacity in real-time, to obtain higher SOC estimation accuracy. When a single cell of the battery pack dies after several tens or hundreds of cycles, The capacity of the battery pack will change greatly, so it will cause a large SOC error. At this time, the capacity correction cannot be carried out, but it can be inferred that a single cell of the battery pack has a fault, and the status of the battery pack can be known in real-time, to provide a reference for the next research battery fault diagnosis.
